



# Composition and Reactivity of Volatile Organic Compounds in the South Coast Air Basin and San Joaquin Valley of California

Shang Liu[1], Barbara Barletta[2], Rebecca S. Hornbrook[3], Alan Fried[4], Jeff Peischl[5,6], Simone Meinardi[2], Matthew Coggon[5,6], Aaron Lamplugh[5,6], Jessica B. Gilman[6], Georgios I. Gkatzelis[5,6,7], Carsten Warneke[6], Eric C. Apel[3], Alan J. Hills[3], Ilann Bourgeois[5,6], James Walega[4], Petter Weibring[4], Dirk Richter[4], Toshihiro Kuwayama[1], Michael FitzGibbon[1], Donald Blake[2]

[1]Research Division, California Air Resources Board, Sacramento, 95814, USA
[2]Department of Chemistry, University of California, Irvine, 92697, USA
[3]Atmospheric Chemistry Observations & Modeling Laboratory, National Center for Atmospheric Research, Boulder, 80301, USA
[4]Institute of Arctic & Alpine Research, University of Colorado, Boulder, 80303 USA
[5]Cooperative Institute for Research in Environmental Sciences, University of Colorado, Boulder 80309, USA
[6]NOAA Chemical Science Laboratory, Boulder, 80305, USA
[7]Now at: Institute of Energy and Climate Research, IEK-8: Troposphere, Forschungszentrum Jülich GmbH, Jülich, 52428, Germany

*Correspondence to*: Shang Liu (shang.liu@arb.ca.gov)

**Abstract.** Comprehensive aircraft measurements of volatile organic compounds (VOCs) covering the South Coast Air Basin (SoCAB) and San Joaquin Valley (SJV) of California were obtained in the summer of 2019. Combined with the CO, $CH_4$, and $NO_x$ data, the total measured gas-phase hydroxyl radical reactivity ($OHR_{TOTAL}$) was quantified. VOCs accounted for ~60%−70% of the $OHR_{TOTAL}$ in both basins. In particular, oxygenated VOCs (OVOCs) contributed > 60% of the OHR of total VOCs ($OHR_{VOC}$) as well as the total observed VOC mixing ratio. Primary biogenic VOCs (BVOCs) represented a minor fraction (< 2%) of the total VOC mixing ratio but accounted for 21% and 6% of the $OHR_{VOC}$ in the SoCAB and SJV, respectively. Furthermore, the contribution of BVOCs to the $OHR_{VOC}$ increased with increasing $OHR_{VOC}$ in the SoCAB, suggesting that BVOCs was important ozone precursors during high ozone episodes. Spatially, the trace gases were heterogeneously distributed in the SoCAB with their mixing ratios and OHR significantly greater over the inland regions than the coast, while their levels were more evenly distributed in the SJV. The results highlight that a better grasp of the emission rates and sources of OVOCs and BVOCs is essential for a predictive understanding of the ozone abundance and distribution in California.

## 1 Introduction

Ambient ozone ($O_3$) is a criteria pollutant that forms from complex photochemical reactions of volatile organic compounds (VOCs) and oxides of nitrogen ($NO_x = NO + NO_2$) in the presence of sunlight. Exposure to $O_3$ can lead to adverse health problems such as airway inflammation and decreased lung function, especially in susceptible populations such as children





(Nuvolone et al., 2017; Kampa and Castanas, 2008). Persistent $NO_x$ emission control measures in California have resulted in

the substantially reduced ambient $O_3$ levels, e.g., the ozone design value (ODV) decreased by over 60% in California's South Coast Air Basin (SoCAB) from 273 parts per billion (ppb) in 1980 to 102 ppb in 2015 (Parrish et al., 2017). However, the SoCAB and San Joaquin Valley (SJV) continue to exceed the health-based, 8-hour National Ambient Air Quality Standard (NAAQS) of 70 ppb particularly during the warm summer months (Faloona et al., 2020; Parrish et al., 2017). The plateauing of $O_3$ levels since 2010 (Wu et al., 2022) poses additional challenges for California's air pollution reduction pathways.


Further reducing ambient $O_3$ may be more difficult than expected due to the nonlinear $O_3$ production rate in response to decreases of $NO_x$ and VOCs and the relatively high $O_3$ background (Parrish et al., 2017). Compared to $NO_x$, VOCs are more complex in their emission sources, composition, and reactivity. VOCs are emitted to the atmosphere from a variety of anthropogenic, pyrogenic, and biogenic sources (Goldstein and Galbally, 2007). After emission, VOCs undergo complex

photochemical reactions resulting in the production of $O_3$. VOCs are comprised of hundreds of molecules with different functionalities. The composition of VOCs in California has been rapidly changing over the past years (Warneke et al., 2012), due in large part to the aggressive emission control measures implemented for on-road mobile sources. This emission source of VOCs is expected to decrease in California over the next decades, driven by the Governor's Executive Order (N-79-20) with a goal of 100% Zero Emission Vehicle sales of new passenger cars and light-duty trucks by 2035. In contrast, the

relative contributions of industrial activities and consumer products to ambient VOCs are growing (Mcdonald et al., 2018; Kim et al., 2022). With a clear path for $NO_x$ reduction in the state, it is critical that the current sources of VOCs and their potential for $O_3$ formation are better characterized to guide future policies that effectively reduce ambient $O_3$.

In this study, we analyzed an extensive set of VOCs and other key trace gases measured from the National Aeronautics and

Space Administration (NASA) DC-8 research aircraft that flew over the SoCAB and SJV in the summer of 2019. We characterized the composition and the source signatures of the VOCs in these two basins. By examining the hydroxyl radical (OH) reactivity of VOCs ($OHR_{VOC}$), we further determined the key VOC species that are responsible for $O_3$ production. The spatial distributions of the $OHR_{VOC}$ in the SoCAB and SJV are also investigated and compared.

## 2 Methods and data processing

The study was conducted in partnership with the Fire Influence on Regional to Global Environments and Air Quality (FIREX-AQ) campaign in the summer of 2019 (Warneke et al., in preparation). We used data from the two California flights, which were designed to study the spatial distribution of VOCs from non-fire emissions over the SoCAB and SJV in California. The measurements were carried out during ~11:00−18:00 (local time) on board the NASA DC-8 research aircraft on 22 July and 5 September 2019, both of which were weekdays. The flights covered the SJV by flying in a raster pattern. In

the SoCAB, the measurements were made over the northern and inland regions of the SoCAB, where the $O_3$ peak events



usually occur (Cai et al., 2019). The flights had fewer measurements in the coastal areas and no measurements over central Los Angeles due to flight restrictions (Fig. 1).

**Figure 1:** Spatial distribution of (a) O₃, (b) NOₓ, (c) VOCs, and (d) OHRᵥₒ꜀. The flight tracks and sample locations are represented by colored lines and points, respectively. The points are sized by flight altitude and colored by the mixing ratios of O₃, NOₓ, VOCs, and OHRᵥₒ꜀ values in (a), (b), (c), and (d), respectively, with the scales shown in the legend. The background map (grey) shows the boundaries of the California air basins, with black indicating the SoCAB and SJV.





**Table 1.** Summary of measurements.

| # | Technique | Species Measured | Sample Duration/ Frequency | Institution | Investigator | Reference |
|---|-----------|------------------|----------------------------|-------------|--------------|-----------|
| 1 | Chemiluminescene (CL) | NO, $NO_2$, $O_3$ | 1 s | NOAA CSL | Tommas Ryerson | Bourgeois et al. (2022); (Bourgeois et al., 2021) |
| 2 | Laser Absorption Spectroscopy (LAS) | CO, $CH_4$ | 1 s | NASA LaRC | Glenn Diskin | Sachse et al. (1987) |
| 3 | Whole Air Sampling (WAS) | $C_2$-$C_{10}$ Alkanes, $C_2$-$C_4$ Alkenes, $C_6$-$C_9$ Aromatics, $C_1$-$C_5$ Alkylnitrates, etc. | 20−100 s, average 40 s, variable frequency | UC Irvine | Donald Blake | Simpson et al. (2020) |
| 4 | HR-ToF-GC/MS | $C_3$-$C_{10}$ hydrocarbons, $C_1$-$C_7$ OVOCs, HCN, $CH_3CN$, $C_1$-$C_2$ halocarbons, etc. | 33 s every 1.75 min | NCAR ACOM | Eric Apel | Apel et al. (2015) |
| 5 | Whole Air Sampling (WAS) | $C_2$-$C_{10}$ Alkanes, $C_2$ $C_4$ Alkenes, $C_6$-$C_9$ Aromatics, $C_1$-$C_5$ Alkylnitrates, etc. | 5 s sample collection every 10-1600 seconds | NOAA CSL | Jessica Gilman | Lerner et al. (2017) |
| 6 | Laser Absorption Spectroscopy (LAS) | HCHO | 1 s | CU Boulder | Alan Fried | Richter et al. (2015), Fried et al. (2020) |
| 7 | $H_3O^+$ ToF-CIMS | Speciated hydrocarbons and OVOCs | 1 s | NOAA CSL | Carsten Warneke | Yuan et al. (2017a) |
| 8 | CIMS | PAN, PPN, other PANs | 1 s | Georgia Tech | Gregory Huey | Zheng et al. (2011) |
| 9 | CIMS | Phenol | 1 s | CalTech | Paul Wennberg | Crounse et al. (2006) |
| 10 | Iodide ToF-CIMS | Formic acid | 1 s | NOAA CSL | Patrick Veres | Veres et al. (2020) |


Here we present analyses of the VOCs, CO, $NO_x$, and $O_3$ measured by eight research groups operating instrumentation on board the DC-8 (Table 1). The technical details of the instruments are described in the Supplementary Materials. Since the diverse techniques have different sampling frequencies, the data were merged to the sampling time (~40 s in duration) of the whole air samples collected by the University of California Irvine (UCI-WAS). The merged data set was created using the

data merge tool provided in the FIREX-AQ data archive (https://www-air.larc.nasa.gov/cgi-bin/ArcView/firexaq). Specifically, the measurements from data set #1 to data set #10 in Table 1 were sequentially merged. The duplicated measurements were removed during each merging step, i.e., if the measurement of a compound existed in the merged data





set and was above detection limit, additional measurements of that compound was not added to the merged data set during the following merging steps.


Fig. S1 shows the comparison of the study-average mixing ratios and standard deviations calculated from the merged data set and the original data sets for the species merged to the UCI-WAS measurements. Each data set was pre-processed such that only the measurements covering the SoCAB and SJV were used for the comparison. The agreement (slope of 1.07 and $r^2$ of 1.00) between the merged data set and the original data set suggests that the merged data set can represent the original

measurements.

To better represent the near-surface mixing ratios, the samples collected at altitudes higher than 4,000 ft (1.2 km) above ground level (a.g.l) were removed from the merged data set. The samples collected over the ocean were also excluded. After screening, the final data set for this study has 69 and 95 samples for the SoCAB and SJV, respectively. A total of 168 gas-

phase compounds were measured, of which 137 had detectable mixing ratio (Table 2). Given that the aircraft speed was about 125 m s$^{-1}$ and the sample collection time was about 40 s, the measurements represented approximately 5-km integrated space along the flight track. The average altitude (± standard deviation) of the samples was 625 (±237) m a.g.l. and 360 (±107) m a.g.l. in the SoCAB and SJV, respectively. Since the samples were collected at least a few hundred meters away from emission sources, the reactive VOCs likely have undergone photochemical processing for minutes to several hours by

the time they were collected. The statistics of the observed mixing ratios of the gas-phase species are summarized in Table 2.

**Table 2.** Statistics of the mixing ratios (median, average, 1-σ standard deviation, and maximum values in ppt unless otherwise specified) and OH reaction rate coefficients of the measured VOCs in the SoCAB and the SJV.

| Compound[a] | SoCAB | | | | SJV | | | | $k_{OH}$[c] | Meas. Tech.[d] |
|---|---|---|---|---|---|---|---|---|---|---|
| | Med.[b] | Avg.[b] | S.D.[b] | Max.[b] | Med.[b] | Avg.[b] | S.D.[b] | Max.[b] | | |
| Alkanes (excluding $CH_4$) | | | | | | | | | | |
| Ethane | 2408 | 2675 | 2041 | 7678 | 1025 | 1282 | 586 | 4052 | 0.25 | WAS |
| Propane | 1041 | 1458 | 1323 | 4927 | 421 | 620 | 429 | 2096 | 1.1 | WAS |
| *n*-Butane | 395 | 415 | 321 | 1247 | 72 | 124 | 130 | 777 | 2.36 | WAS |
| Isobutane | 259 | 259 | 189 | 719 | 45 | 78 | 80 | 378 | 2.1 | WAS |
| *n*-Pentane | 248 | 220 | 167 | 603 | 30 | 54 | 68 | 554 | 6.911 | WAS |
| Isopentane | 530 | 508 | 409 | 1565 | 65 | 103 | 113 | 811 | 3.6 | WAS |
| Cyclopentane | 26 | 22 | 14 | 56 | 6 | 9 | 8 | 37 | 5.0 | WAS |
| *n*-Hexane | 73 | 74 | 58 | 242 | 12 | 18 | 15 | 73 | 5.2 | WAS |
| 2-Methylpentane | 118 | 110 | 80 | 292 | 15 | 23 | 19 | 93 | 5.2 | WAS |
| 3-Methylpentane | 67 | 64 | 50 | 182 | 8 | 13 | 12 | 61 | 5.2 | WAS |
| 2,2-Dimethylbutane | 29 | 28 | 19 | 75 | 5 | 7 | 4 | 23 | 2.2 | WAS |
| 2,3-Dimethylbutane | 40 | 39 | 29 | 116 | 7 | 9 | 6 | 28 | 5.8 | WAS |
| Methylcyclopentane | 69 | 62 | 47 | 189 | 12 | 16 | 13 | 64 | 5.7 | WAS |





| | | | | | | | | | | |
|---|---|---|---|---|---|---|---|---|---|---|
| Cyclohexane | 22 | 25 | 17 | 72 | 6 | 9 | 7 | 32 | 7.0 | WAS |
| n-Heptane | 27 | 30 | 22 | 83 | 7 | 11 | 8 | 37 | 6.8 | WAS |
| 2-Methylhexane | 32 | 33 | 24 | 95 | 6 | 8 | 5 | 26 | 6.9 | WAS |
| 3-Methylhexane | 34 | 36 | 27 | 104 | 7 | 9 | 6 | 27 | 7.2 | WAS |
| 2,3-Dimethylpentane | 43 | 44 | 32 | 124 | 6 | 7 | 4 | 19 | 6.5 | WAS |
| 2,4-Dimethylpentane | 47 | 46 | 27 | 103 | 6 | 8 | 6 | 20 | 4.8 | WAS (NOAA) |
| Methylcyclohexane | 19 | 21 | 14 | 58 | 4 | 8 | 7 | 32 | 9.6 | WAS |
| n-Octane | 13 | 15 | 10 | 46 | 6 | 6 | 2 | 11 | 8.1 | WAS |
| 2,2,4-Trimethylpentane | 55 | 65 | 50 | 196 | 6 | 8 | 6 | 39 | 3.3 | WAS |
| 2,3,4-Trimethylpentane | 21 | 21 | 15 | 61 | 6 | 7 | 4 | 17 | 6.6 | WAS |
| n-Nonane | 9 | 10 | 6 | 34 | 4 | 5 | 1 | 9 | 9.7 | WAS |
| n-Decane | 7 | 9 | 6 | 29 | 4 | 5 | 1 | 7 | 11 | WAS |
| n-Undecane | 4 | 6 | 3 | 16 | 4 | 4 | 1 | 8 | 12 | WAS |
| Alkenes | | | | | | | | | | |
| Ethene | 266 | 312 | 303 | 1181 | 47 | 63 | 14 | 248 | 8.5 | WAS |
| Propene | 38 | 45 | 44 | 181 | 12 | 14 | 8 | 59 | 26 | WAS |
| Propadiene | 6 | 8 | 4 | 16 | NA | NA | NA | NA | 9.8 | WAS |
| 1-Butene | 6 | 8 | 6 | 25 | 4 | 4 | 1 | 7 | 31 | WAS |
| Isobutene | 9 | 11 | 7 | 23 | 9 | 10 | 4 | 15 | 51 | WAS |
| 1,3-Butadiene | 5 | 6 | 2 | 8 | 7 | 7 | 1 | 7 | 67 | WAS |
| trans-2-Butene | 1 | 1 | 1 | 5 | 1 | 2 | 4 | 15 | 64 | WAS (NOAA) |
| 1-Pentene | 4 | 5 | 2 | 8 | 3 | 4 | 1 | 6 | 31 | WAS |
| trans-2-Pentene | 4 | 5 | 2 | 8 | NA | NA | NA | NA | 67 | WAS |
| Cis-2-Pentene | 1 | 1 | 1 | 2 | 0 | 1 | 1 | 2 | 65 | WAS (NOAA) |
| 2-Methyl-1-butene | 5 | 5 | 2 | 8 | 5 | 6 | 3 | 10 | 61 | WAS |
| 3-Methyl-1-butene | 4 | 4 | 1 | 5 | NA | NA | NA | NA | 32 | WAS |
| 2-Methyl-2-butene | 4 | 4 | 1 | 4 | 3 | 3 | 0 | 3 | 87 | WAS |
| 1,3-Pentadiene | NA | NA | NA | NA | 4 | 4 | 1 | 4 | 101 | WAS |
| 1-Hexene | 5 | 6 | 1 | 8 | 6 | 9 | 8 | 23 | 37 | WAS |
| 1-Heptene | 10 | 11 | 8 | 32 | 5 | 7 | 4 | 14 | 40 | WAS |
| 1-Octene | NA | NA | NA | NA | 7 | 7 | 6 | 11 | 30 | WAS |
| 1-Nonene | 3 | 3 | 0 | 4 | NA | NA | NA | NA | 42 | WAS |
| Aromatics | | | | | | | | | | |
| Benzene | 74 | 82 | 65 | 222 | 21 | 25 | 13 | 76 | 1.2 | WAS |
| Toluene | 137 | 160 | 141 | 551 | 17 | 24 | 21 | 131 | 5.6 | WAS |
| Ethylbenzene | 25 | 26 | 18 | 65 | 5 | 7 | 4 | 17 | 7.0 | WAS |
| m/p-Xylene | 42 | 49 | 47 | 173 | 7 | 9 | 7 | 33 | 19 | WAS |
| o-Xylene | 22 | 24 | 19 | 79 | 5 | 6 | 3 | 15 | 14 | WAS |



| | | | | | | | | | |
|---|---|---|---|---|---|---|---|---|---|
| Styrene | 6 | 8 | 7 | 29 | 4 | 4 | 1 | 4 | 58 | WAS |
| Isopropylbenzene | 4 | 4 | 1 | 6 | NA | NA | NA | NA | 6.3 | WAS |
| *n*-Propylbenzene | 5 | 6 | 2 | 11 | 3 | 4 | 1 | 5 | 5.8 | WAS |
| 2-Ethyltoluene | 4 | 6 | 3 | 13 | 4 | 4 | 1 | 5 | 12 | WAS |
| 3-Ethyltoluene | 6 | 8 | 7 | 24 | 3 | 4 | 1 | 5 | 19 | WAS |
| 4-Ethyltoluene | 5 | 6 | 4 | 17 | 4 | 4 | 1 | 7 | 12 | WAS |
| 1,3,5-Trimethylbenzene | 4 | 4 | 1 | 5 | NA | NA | NA | NA | 57 | WAS |
| 1,2,4-Trimethylbenzene | 6 | 9 | 8 | 27 | 5 | 5 | 2 | 8 | 33 | WAS |
| Naphthalene | 10 | 10 | 6 | 30 | 9 | 9 | 3 | 17 | 23 | $H_3O^+$ ToF-CIMS |
| **BVOCs and Related Oxidation Products** | | | | | | | | | | |
| Methacrolein (MAC) | 126 | 196 | 193 | 692 | 31 | 41 | 33 | 156 | 29 | WAS |
| Methyl vinyl ketone (MVK) | 166 | 303 | 317 | 1292 | 120 | 134 | 88 | 474 | 20 | WAS |
| Isoprene | 156 | 178 | 178 | 651 | 14 | 36 | 55 | 298 | 100 | WAS |
| α-Pinene | 6 | 8 | 5 | 20 | 6 | 6 | 3 | 11 | 53 | WAS |
| β-Pinene | 5 | 4 | 1 | 6 | NA | NA | NA | NA | 74 | WAS |
| **OVOCs** | | | | | | | | | | |
| Methanol | 7734 | 7703 | 3858 | 15980 | 16349 | 17062 | 4350 | 29719 | 0.94 | HR-ToF-GC/MS |
| Formaldehyde | 4042 | 4475 | 3021 | 9503 | 3703 | 4239 | 1507 | 10067 | 9.4 | LAS (CU Boulder) |
| Formic acid | 5444 | 4344 | 2861 | 8506 | 9837 | 9702 | 1872 | 13211 | 0.45 | CIMS (NOAA) |
| Carbonyl sulfide | 616 | 611 | 38 | 678 | 611 | 615 | 26 | 702 | 0.0020 | WAS |
| Nitromethane | 40 | 61 | 63 | 360 | 81 | 105 | 76 | 422 | 0.016 | WAS |
| Methyl nitrate | 9 | 10 | 4 | 17 | 10 | 10 | 1 | 14 | 0.023 | WAS |
| Ethyl nitrate | 4 | 4 | 2 | 9 | 3 | 4 | 1 | 6 | 0.18 | WAS |
| Isocyanic acid | 238 | 354 | 269 | 1151 | 210 | 337 | 290 | 1104 | 0.0012 | CIMS (NOAA) |
| Ethanol | 793 | 1804 | 2500 | 10031 | 2590 | 2933 | 2020 | 11019 | 3.2 | WAS |
| Acetaldehyde | 1182 | 1307 | 914 | 3229 | 1229 | 1516 | 797 | 4410 | 15 | HR-ToF-GC/MS |
| Methyl formate | 151 | 148 | 52 | 226 | 172 | 177 | 24 | 209 | 0.22 | HR-ToF-GC/MS |
| PAN | 974 | 914 | 812 | 3220 | 546 | 668 | 393 | 2463 | 0.042 | CIMS (Georgia Tech) |
| Isopropanol | 104 | 153 | 187 | 854 | 55 | 60 | 28 | 187 | 5.1 | WAS |
| Propanal | 144 | 157 | 105 | 434 | 122 | 138 | 54 | 317 | 20 | HR-ToF-GC/MS |
| Acrolein | 44 | 58 | 61 | 434 | 50 | 53 | 31 | 186 | 20 | WAS |
| Acetone | 4285 | 4415 | 2655 | 9953 | 2896 | 3176 | 760 | 5649 | 0.17 | HR-ToF-GC/MS |
| *n*-Propyl nitrate | 1 | 2 | 1 | 5 | 1 | 1 | 0 | 2 | 0.58 | WAS |



| | | | | | | | | | | |
|---|---|---|---|---|---|---|---|---|---|---|
| Isopropyl nitrate | 11 | 12 | 9 | 35 | 7 | 8 | 3 | 17 | 0.29 | WAS |
| PPN | 100 | 96 | 95 | 475 | 49 | 62 | 40 | 215 | 0.46 | CIMS (Georgia Tech) |
| APAN | 17 | 28 | 31 | 131 | 20 | 21 | 10 | 58 | 16 | CIMS (Georgia Tech) |
| Carbon suboxide | 6 | 6 | 2 | 9 | 5 | 5 | 1 | 7 | 2.6 | HR-ToF-GC/MS |
| Methyl acetate | 68 | 95 | 84 | 404 | 74 | 78 | 22 | 139 | 0.26 | WAS |
| Butanal | 37 | 40 | 25 | 145 | 36 | 42 | 21 | 120 | 24 | WAS |
| Isobutanal | 18 | 21 | 12 | 58 | 18 | 19 | 8 | 58 | 26 | WAS |
| Crotonaldehyde | 13 | 15 | 9 | 34 | 15 | 15 | 2 | 21 | 36 | HR-ToF-GC/MS |
| Methyl ethyl ketone (MEK) | 185 | 250 | 199 | 898 | 140 | 186 | 116 | 545 | 1.1 | WAS |
| 2,3-Butanedione | 16 | 16 | 9 | 44 | 24 | 24 | 6 | 44 | 0.25 | HR-ToF-GC/MS |
| Furan | NA | NA | NA | NA | 8 | 8 | 7 | 13 | 40 | WAS |
| 2-Furanone | 39 | 36 | 20 | 74 | 75 | 70 | 25 | 111 | 45 | PTR-MS |
| Tetrahydrofuran | 5 | 6 | 4 | 19 | 1 | 2 | 1 | 3 | 18 | HR-ToF-GC/MS |
| Ethyl acetate | 23 | 25 | 19 | 85 | 20 | 33 | 37 | 235 | 1.63 | HR-ToF-GC/MS |
| Methyl propionate | 2 | 2 | 1 | 3 | 2 | 2 | 1 | 5 | 0.88 | HR-ToF-GC/MS |
| Maleic anhydride | 40 | 37 | 24 | 82 | 18 | 23 | 10 | 50 | 1.5 | $H_3O^+$ ToF-CIMS |
| Isobutyl nitrate | 19 | 21 | 11 | 40 | 13 | 12 | 5 | 22 | 1.5 | WAS and HR-ToF-GC/MS |
| 2-Butyl nitrate | 10 | 12 | 10 | 38 | 5 | 6 | 3 | 22 | 0.86 | WAS |
| PBN | 25 | 28 | 23 | 103 | 11 | 14 | 10 | 72 | 4.7 | CIMS (Georgia Tech) |
| 2-Pentyl nitrate | 3 | 4 | 3 | 12 | 1 | 2 | 1 | 7 | 1.7 | WAS |
| 3-Pentyl nitrate | 2 | 2 | 2 | 7 | 1 | 1 | 1 | 4 | 1.0 | WAS |
| 3-Methyl-2-butyl nitrate | 4 | 4 | 4 | 17 | 1 | 2 | 1 | 6 | 1.7 | WAS |
| Phenol | 6 | 6 | 4 | 16 | 4 | 7 | 6 | 21 | 27 | CIMS (CalTech) |
| Catechol/5-Methylfurfural | 45 | 38 | 23 | 68 | 37 | 37 | 12 | 60 | 78 | $H_3O^+$ ToF-CIMS |
| Guaiacol | 14 | 15 | 11 | 32 | 19 | 19 | 5 | 25 | 54 | $H_3O^+$ ToF-CIMS |
| Benzaldehyde | 30 | 29 | 21 | 72 | 27 | 30 | 17 | 80 | 12 | $H_3O^+$ ToF-CIMS |
| Creosol | 12 | 11 | 7 | 25 | 10 | 11 | 3 | 17 | 75 | $H_3O^+$ ToF-CIMS |



| Syringol | 4 | 5 | 2 | 9 | 4 | 4 | 2 | 9 | 81 | $H_3O^+$ ToF-CIMS |
|---|---|---|---|---|---|---|---|---|---|---|
| | | | | | Other VOCs | | | | | |
| Chloromethane | 572 | 567 | 36 | 651 | 561 | 569 | 33 | 679 | 0.036 | WAS |
| Dichloromethane | 80 | 99 | 73 | 450 | 69 | 68 | 5 | 80 | 0.12 | WAS |
| Chloroform | 21 | 24 | 13 | 57 | 18 | 19 | 3 | 29 | 0.11 | WAS |
| Tetrachloromethane | 78 | 78 | 1 | 81 | 79 | 79 | 1 | 82 | 0.00 | WAS |
| Bromomethane | 7 | 8 | 2 | 15 | 11 | 12 | 9 | 89 | 0.029 | WAS |
| Dibromomethane | 1 | 1 | 0 | 1 | 1 | 1 | 0 | 1 | 0.11 | WAS |
| Bromoform | 2 | 2 | 1 | 5 | 1 | 1 | 0 | 1 | 0.15 | WAS |
| Dibromochloromethane | 3 | 4 | 3 | 12 | 1 | 1 | 0 | 2 | 0.22 | WAS |
| Bromodichloromethane | 2 | 2 | 2 | 6 | 1 | 1 | 0 | 2 | 1.2 | WAS |
| Iodomethane | 2 | 1 | 1 | 4 | 1 | 1 | 0 | 2 | 0.10 | WAS |
| Methanethiol | 9 | 9 | 6 | 21 | 11 | 10 | 3 | 18 | 33 | HR-ToF-GC/MS |
| Hydrogen cyanide | 209 | 246 | 85 | 457 | 195 | 212 | 42 | 321 | 0.03 | CIMS (CalTech) |
| Ethyne | 284 | 342 | 308 | 1185 | 87 | 104 | 47 | 318 | 0.75 | WAS |
| Chloroethane | 3 | 3 | 1 | 6 | 3 | 3 | 1 | 6 | 0.40 | WAS |
| 1,2-Dichloroethane | 16 | 16 | 6 | 30 | 17 | 17 | 1 | 20 | 0.23 | WAS |
| Methyl chloroform | 2 | 2 | 0 | 3 | 2 | 2 | 0 | 2 | 0.0095 | WAS |
| Trichloroethene | 2 | 4 | 4 | 17 | 0 | 1 | 0 | 3 | 1.9 | WAS |
| Tetrachloroethene | 5 | 11 | 10 | 37 | 2 | 3 | 4 | 39 | 0.16 | WAS |
| Dimethyl sulfide | 4 | 5 | 3 | 16 | 5 | 8 | 10 | 63 | 7.0 | WAS |
| Carbon disulfide | 5 | 5 | 3 | 12 | 5 | 5 | 1 | 8 | 1.2 | HR-ToF-GC/MS |
| Acetonitrile | 169 | 201 | 76 | 438 | 160 | 166 | 23 | 240 | 0.022 | HR-ToF-GC/MS |
| Propyne | 13 | 14 | 7 | 35 | 5 | 5 | 1 | 8 | 0.94 | WAS |
| Acrylonitrile | 12 | 14 | 6 | 22 | NA | NA | NA | NA | 4.0 | WAS |
| Propanenitrile | 13 | 15 | 8 | 36 | 14 | 15 | 5 | 29 | 0.14 | WAS |
| D5 | 21 | 18 | 13 | 40 | 5 | 5 | 3 | 12 | 2.1 | HR-ToF-GC/MS |
| | | | | | $NO$, $NO_2$, CO, and $CH_4$ | | | | | |
| NO | 229 | 747 | 1276 | 5921 | 138 | 154 | 75 | 426 | 10 | CL |
| $NO_2$ | 1425 | 2997 | 3661 | 17891 | 879 | 976 | 514 | 2691 | 12 | CL |
| CO (ppb) | 196 | 193 | 92 | 354 | 130 | 137 | 18 | 206 | 0.24 | LAS (NASA) |
| $CH_4$ (ppb) | 1946 | 1957 | 71 | 2091 | 2070 | 2158 | 239 | 3437 | 0.0064 | LAS (NASA) |

[a]The following species were measured but not reported as their mixing ratios were below detection limit: *cis*-2-butene, 1,2-butadiene, 1-buten-3-yne, 1,3-butadiyne, 1-butyne, 2-butyne, *cis*-2-pentene, 3-methyl-1-pentene/4-methyl-1-pentene, 1-decene, cyclopentene, ethynylbenzene, chlorobenzene, tricyclene, camphene, myrcene, limonene, Δ3-carene, 2-methylfuran,





3-methylfuran, benzofuran, 1,3-butadiyne, chloroiodomethane, 2-methyl-3-buten-2-ol, methylacrylonitrile, pyrrole, 2-ethylfuran, dimethylfurans, vinylfuran, 3-furaldehyde, and 1,3-pentadiene.

The mixing ratios of carbon suboxide, 2,3-butanedione, tetrahydrofuran, ethyl acetate, methyl propionate, and methanthiol from the HR-ToF-GC/MS measurement was estimated using relative sensitivities. The sum of these species accounted for less than 0.2% of the total VOC mixing ratio in the SoCAB and SJV.

The following abbreviations are used: D5 represents decamethylcyclopentasiloxane, PAN represents peroxyacetyl nitrate, PPN represents peroxylpropionyl nitrate, APAN represents peroxyacryloyl nitrate, and PBN represents peroxybutyryl nitrate.

[b]NA represents not applicable because the VOCs are below detection limit.

[c]$k_{OH}$ in $10^{-12}$ $cm^3$ molecule$^{-1}$ s$^{-1}$.

[d]WAS refers to the UC Irvine WAS unless otherwise stated.

The $H_3O^+$ ToF-CIMS measurements and the formic acid mixing ratio (from iodide ToF-CIMS) were only available during the flight on 5 September 2019.

Hydrogen cyanide was measured by the CalTech CIMS and HR-ToF-GC/MS. The high time resolution (1 Hz) CalTech CIMS measurement was used.

The mixing ratio of isobutyl nitrate was derived by the difference between the sum of isobutyl nitrate and 2-butyl nitrate (measured by the HR-ToF-GC/MS) and 2-butyl nitrate (measured by the UC Irvine WAS).

## 3 Results and discussion

### 3.1 Mixing ratio and spatial distribution of $O_3$, VOCs, and $NO_x$

The study-average aircraft-measured $O_3$ (± 1 σ) mixing ratio was 70.2±26.7 ppb in the SoCAB and 73.5±9.5 ppb in the SJV. The maximum observed $O_3$ mixing ratio was 122.0 ppb and 101.0 ppb in the SoCAB and SJV, respectively. The average non-$CH_4$ VOC mixing ratio (± 1 σ) in the SoCAB and SJV was 36.9.0±7.3 ppb and 45.8±5.6 ppb, respectively (for simplicity, we use VOC to represent non-$CH_4$ VOC hereafter). The higher VOC levels in the SJV were mainly driven by methanol. The average observed methanol was 17.1 ppb in the SJV, which doubled the average 7.7 ppb observed in the SoCAB. The average aircraft-measured $NO_x$ (± 1 σ) was 3.8±4.8 ppb in the SoCAB, which was 3.5 times the $NO_x$ (± 1 σ) of 1.1±0.6 ppb in the SJV. The high $NO_x$ levels in the SoCAB resulted in a relatively lower VOC-to-$NO_x$ ratio of 9.7±12.4 compared to 41.6±23.3 in the SJV.

The spatial distributions of $O_3$, VOCs, and $NO_x$ are shown in Fig. 1 (a-c). In the SoCAB, the mixing ratios of $O_3$, VOCs, and $NO_x$ generally increased from coastal to inland regions, with the highest mixing ratios occurring near the northern boundary of the SoCAB. The coast-inland gradient likely resulted from the accumulation and chemical aging of the air pollutants as they were transported eastwards. In the SJV, the mixing ratios of $O_3$, VOCs, and $NO_x$ were more homogeneously distributed, i.e., the occurance frequency of the high mixing ratios was smaller compared to that in the SoCAB (Figs. S2-S4). Since the very high mixing ratios determine the design values (Parrish et al., 2017), the result suggests that $O_3$ in the SoCAB is more



prone to exceed the NAAQS. This is consistent with the higher $O_3$ design values in the SoCAB compared to the SJV resulted from analyses using the monitoring network data (Parrish et al., 2017).

**3.2 Composition and source signatures of VOCs**

The measured VOCs are grouped into chemical families based on their characteristic functional groups or emission sources
(for biogenic VOCs and their oxidation products). The average VOC composition from the two research flights was similar, so the study-average composition (i.e., average of all samples collected during the two flights) is presented (Fig. 2). The mean, median, and maximum mixing ratios for each VOC measured in this study are listed in Table 2. Oxygenated VOCs (OVOCs) and alkanes dominated the measured VOC mixing ratios, accounting for 91%−96% of the total measured VOC mixing ratio in both the SoCAB and SJV. This is comparable to the measurements conducted at Pasadena in 2010 during the
California Research at the Nexus of Air Quality and Climate Change (CalNex) field study, where OVOCs and alkanes comprised ~80% of the measured reactive organic carbon (ROC) mass (Heald et al., 2020). They are also the two most abundant chemical families in other regions of the world, e.g., in Seoul, South Korea (Simpson et al., 2020; Kim et al., 2018) and Hong Kong (Ling et al., 2014). Specifically, OVOCs accounted for 74% and 91% of the total measured VOC mixing ratio in the SoCAB and SJV, respectively, and the contributions of alkanes were 17% and 5% in the SoCAB and SJV,
respectively. In contrast, primary biogenic VOCs (BVOCs) are a minor fraction (< 2%) of the total measured VOCs mixing ratio in both basins.

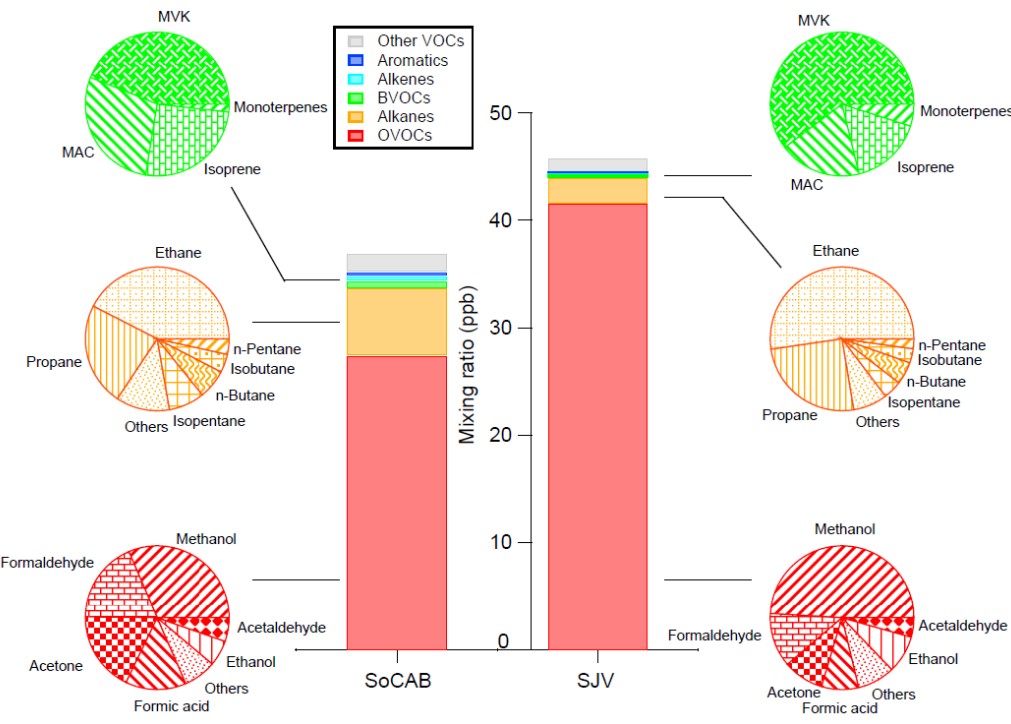

**Figure 2:** The campaign-average VOC composition for the SoCAB and SJV.



Methanol was the most abundant OVOC in both basins, which was responsible for 28% and 41% of the OVOC mixing ratio in the SoCAB and SJV, respectively. Methanol in the SoCAB (7.7 ppb) was about half of the methanol levels in the SJV (17.1 ppb). Very high methanol (> 25 ppb) was observed over the southeast (e.g., Delano, Tulare, Visalia) regions of Fresno, where dense, confined animal farming operations (CAFOs) are located, suggesting direct methanol emissions from dairy operations. Similar to methanol, formic acid in the SoCAB (4.3 ppb) was about half of the formic acid in the SJV (9.7 ppb).

Formic acid has been identified as a major fatty acid from dariy operations (Page et al., 2014; Mårtensson et al., 1999). Formic acid also increased with methanol ($r^2$ = 0.7), suggesting that a dairies may be a major source of formic acid in the SJV. The region with high methanol mixing ratio also features high ethanol levels, which likely resulted from dairy emissions as well. Studies have shown that methanol and ethanol are the dominant VOCs emitted from dairy cows and their wastes (Gentner et al., 2014; Shaw et al., 2007; Sun et al., 2008). In addition, ethanol is likely the most abundant VOC

species emitted from dairy silages in the SJV (Malkina et al., 2011; Yuan et al., 2017b), while consumer product use has a sizeable contribution to its ambient mixing ratios in urban cores (Mcdonald et al., 2018; Coggon et al., 2021). Vehicle operations also emit ethanol into the atmosphere due to the increasing use of the E10 and E85 ethanol-gasoline blends (De Gouw et al., 2012; Gkatzelis et al., 2021). More detailed surface-level source apportionment methods must be employed to determine the specificity of its origin. The high ethanol levels in the SJV likely led to high mixing ratios of its oxidation

product, e.g., acetaldehyde, which was 16% higher in the SJV than that in the SoCAB.

Formaldehyde (HCHO) was the second and third most abundant OVOC, with comparable mixing ratio (~4.5 ppb) in the SoCAB and SJV, respectively. HCHO contributed to 16% and 10% to the OVOC mixing ratio in the SoCAB and SJV, respectively. Primary sources of HCHO include automotive exhaust (Anderson et al., 1996), industrial emissions

(Salthammer et al., 2010), biomass burning (Holzinger et al., 1999), and off-gassing from building materials (Gilbert et al., 2008; Liu et al., 2016). HCHO can also form from the photooxidation of VOCs, such as alkenes and BVOCs (Choi et al., 2010; Parrish et al., 2012). Given the contrast between anthropogenic emission sources in the SoCAB and SJV (e.g., the CO emissions in the SoCAB are 1.4 times the CO emissions in the SJV, estimated from California Air Resources Board's emission inventory), the similarity of HCHO mixing ratio between the two regions during the two flight days suggests that

photooxidation of non-anthropogenic emissions (e.g., BVOCs) was likely the dominant regional driver of HCHO, which is consistent with the findings from previous studies in North America (Lee et al., 1998; Palmer et al., 2003). The high methanol, ethanol, and formic acid mixing ratios in the SJV made the OVOC level in the SJV 52% higher compared to the OVOC level in the SoCAB. Acetone also had sizable contributions to the OVOC mixing ratio (Fig. 2).

The composition of alkanes in the SoCAB and SJV was similar, but the total alkane mixing ratio was 2.6 times higher in the SoCAB. Ethane and propane were the dominant species. Specifically, ethane accounted for ~40%−50% of the total alkane mixing ratio, followed by propane (~25%) in both regions. The linear regressions of propane versus ethane for the SoCAB



and SJV fell on the same line, with a Pearson's *r*-value of 0.96 and a slope of 0.6. Peischl et al. (2013) summarized the propane versus ethane slope values from different sources. The slope obtained from this study is consistent with the aircraft

observation over the SoCAB in 2010 and fell within the slope range for local oil and gas well emissions (Peischl et al., 2013). In contrast, the propane-to-ethane ratios for pipeline quality natural gas are ~0.15 as heavier alkanes are removed during processing (Wennberg et al., 2012; Peischl et al., 2013). On-road emission ratios of propane-to-ethane are also very low, but relatively minor as gasoline and diesel do not contain large amounts of these alkanes (Fraser et al., 1998). This result suggests that oil and gas production activities are likely the major source of propane and ethane in the SoCAB and

SJV. Isopentane and *n*-pentane were tightly correlated in both basins but with different slopes. The isopentane/*n*-pentane ratio was 2.4 in the SoCAB. This value is consistent with ratios of 2.3–3.8 observed for regions highly impacted by vehicular emissions (Gilman et al., 2013). In contrast, the isopentane/*n*-pentane ratio was 1.8 for SJV, suggesting that the air mass was mixed with non-urban emissions.

The BVOCs accounted for < 2% of the total VOC mixing ratio in both basins. In the BVOC category, isoprene is assumed to be only from biogenic emissions and anthropogenically-driven isoprene emissions are negligible in summer (Guenther et al., 2012; Reimann et al., 2000). The BVOCs were dominated by isoprene and its oxidation products methacrolein (MAC) and methyl vinyl ketone (MVK) (Wennberg et al., 2018), the sum of which accounted for ~95% of the total observed BVOC mixing ratio. MVK was the most abundant BVOC species that contributed ~50% of BVOC mixing ratio, followed by nearly

equal contributions of isoprene and MAC. The monoterpenes (such as α- and β-pinene) were minor, with their average mixing ratios smaller than 10 ppt.

### 3.3 Total OH reactivity

Quantifying the OH reactivity (OHR) of reactive gases provides estimates of the potential roles of individual $O_3$ precursors (Ling et al., 2014; Gilman et al., 2009; Mcdonald et al., 2018). The OHR of a compound is calculated from the compound

concentration multiplied by its reaction rate coefficient with the OH radical. The total OHR is the sum of the OHR of all reactants in the atmosphere, which is given by the following equation:

$$\begin{aligned} \text{OHR}_{\text{TOTAL}} = {} & \text{OHR}_{\text{CH}_4} + \text{OHR}_{\text{CO}} + \text{OHR}_{\text{NO}_x} + \text{OHR}_{\text{VOC}} \\ = {} & k_{\text{OH+CH}_4}[\text{CH}_4] + k_{\text{OH+CO}}[\text{CO}] + k_{\text{OH+NO}_x}[\text{NO}_x] + \sum(k_{\text{OH+VOC}}[\text{VOC}]) \quad (1) \end{aligned}$$


where the subscript of OHR indicates the species X used to calculate the OHR, [X] is the mixing ratio of X, and $k_{\text{OH+X}}$ is the reaction rate coefficient of X with the OH radical. The $k_{\text{OH+X}}$ values were obtained from literature or kinetic databases (Atkinson and Arey, 2003; Atkinson et al., 2004; Atkinson et al., 2008; Kwok and Atkinson, 1995; Burkholder et al., 2015; Alton and Browne, 2020; Atkinson, 1986; Atkinson et al., 2006; Atkinson et al., 1997; Bierbach et al., 1995; Bierbach et al.,





1994; Borduas et al., 2016; Lauraguais et al., 2015; Roberts et al., 2003; Semadeni et al., 1995) (National Institute of Standards and Technology Chemical Kinetics Database) and are tabulated in Table 2. We used the rate constants at 298 K. This is reasonable as the average ambient air temperature (± standard deviation) of the samples was 26.7(±2.1) °C and 31.0(±1.6) °C in the SoCAB and SJV, respectively. We note that the $OHR_{VOC}$ was calculated using the measured VOCs. There is likely missing OHR from unmeasured VOCs due to the limitation of the quantification methods. Previous studies

show that the fraction of the missing OHR ranges from 5% to 80%, depending on the sampling location, time, and the number of measured VOC species (Yang et al., 2016; Hansen et al., 2021). With the extensive VOC measurements in this study, it is expected that the fraction of the missing OHR is close to the lower end of the 5%−80% range. Nevertheless, the $OHR_{VOC}$ reported in this study should be interpreted as a minimum for the OHR of the total VOCs.

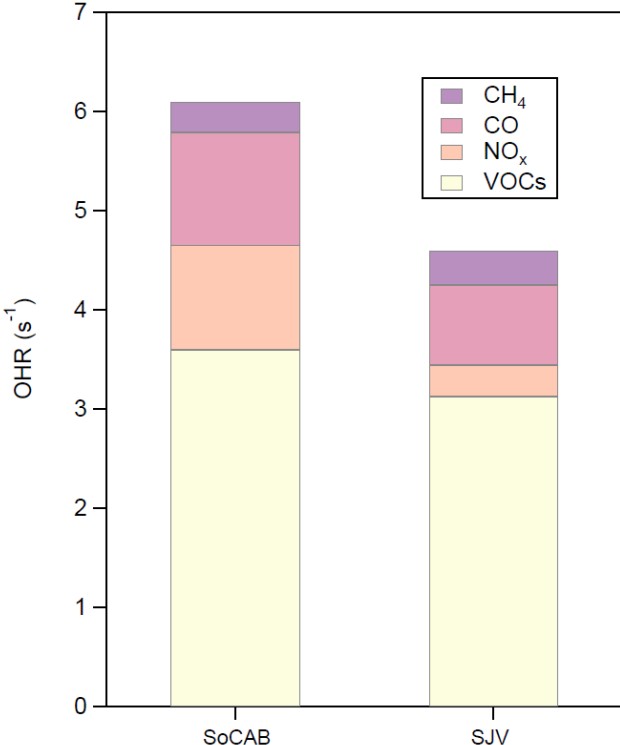

**Figure 3:** The total measured OHR for the SoCAB and SJV.

The calculated $OHR_{TOTAL}$ was 6.1 s$^{-1}$ and 4.6 s$^{-1}$ for the SoCAB and SJV, respectively (Fig. 3). The O$_3$ mixing ratio generally increased nonlinearly with increasing $OHR_{VOC}$, $OHR_{CO}$, $OHR_{CH4}$, and $OHR_{NOx}$ in both basins, especially for O$_3$ > 40 ppb (Fig. 4). The $OHR_{TOTAL}$ values observed in this study are generally less than half of the $OHR_{TOTAL}$ reported in Pasadena and Central California 10–20 years ago (Table S1). However, we note that the comparison may be biased as the previous studies

are from ground measurements. The percentage contributions of CH$_4$, CO, NO$_x$, and VOCs to $OHR_{TOTAL}$ were similar between the SoCAB and SJV, with the VOCs accounting for ~60% of the $OHR_{TOTAL}$ (Fig. 3). For reference, the global annual mean contribution of VOCs to $OHR_{TOTAL}$ is ~50% from a recent review study (Heald and Kroll, 2020). The large



contribution of VOCs to $OHR_{TOTAL}$ is consistent with previous measurements in California (Table S1), suggesting that VOCs remain a high potential for $O_3$ formation in the SoCAB and SJV. CO was the second most important species following VOCs, and was responsible for ~18% of the $OHR_{TOTAL}$ in both basins. The contribution of $NO_x$ to $OHR_{TOTAL}$ was 17% in the SoCAB and 6% in the SJV.

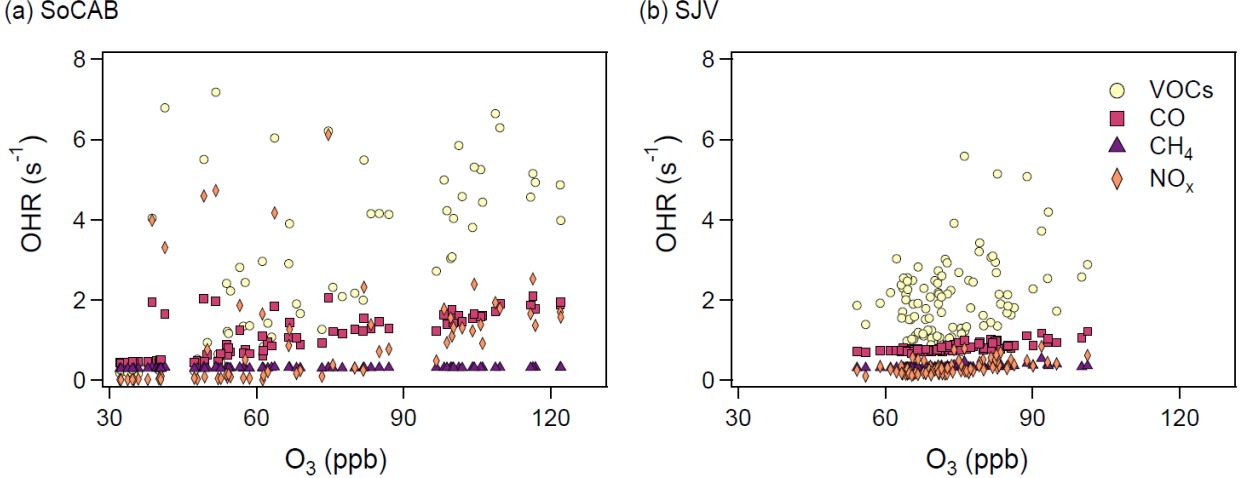

**Figure 4:** Scatter plot of $OHR_{VOC}$, $OHR_{CO}$, $OHR_{CH4}$, and $OHR_{NOx}$ versus $O_3$ for the (a) SoCAB and (b) SJV.

### 3.4 OH reactivity of individual VOCs

The OHR of the individual species for the SoCAB and SJV generally follows similar order. Linear regression of the $OHR_{SOCAB}$ versus the $OHR_{SJV}$ for the individual species shows a slope of 1.3 and $r^2$ of 0.8 (Fig. 5), suggesting that the overall OHR in the SoCAB was 30% higher than that in the SJV. The top five VOCs that have the largest OHR in the SoCAB were HCHO, acetaldehyde, isoprene, methanol, and MVK in decreasing order. In the SJV, the top five most abundant VOCs in terms of OHR were formaldehyde, acetaldehyde, methanol, ethanol, and formic acid. The sum of these species accounted for 63% and 73% of the calculated $OHR_{VOC}$ in the SoCAB and SJV, respectively. These results suggest that (i) aldehydes are the key compounds contributing to OHR in both basins, (ii) biogenic VOCs are important for OHR in the SoCAB, and (iii) dairy emissions are likely important for OHR in the SJV. Compared to the OHR of the VOCs measured in Pasadena in 2010 during the CalNex campaign (57 species measured in common), the majority of the species measured in the SoCAB (52 out of 57) in 2019 had smaller OHR values, with a mean percentage decrease of 53%.





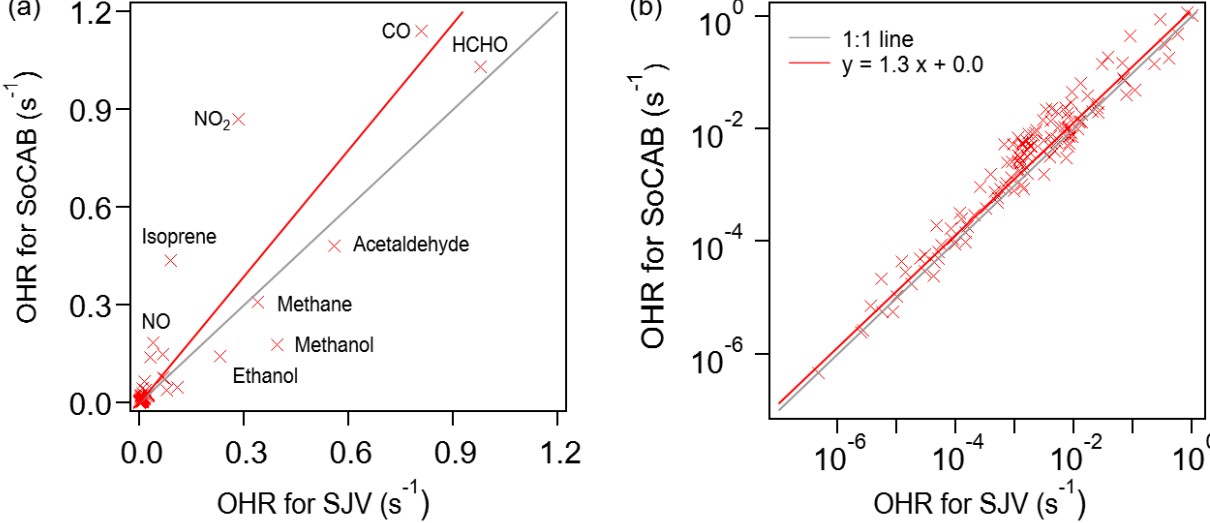

**Figure 5:** Scatter plot of OHR$_{SoCAB}$ versus OHR$_{SJV}$ for the individual species in (a) linear scale and (b) logarithmic scale. The red line represents linear regression using the orthogonal distance regression method.

The OVOCs category was the largest contributor to the OHR$_{VOC}$ among all the categories (Fig. 6; Table S2). The OHR$_{OVOC}$ was 3.6 s$^{-1}$ in the SoCAB and 3.1 s$^{-1}$ in the SJV, which was responsible for 59% and 68% of the OHR$_{VOC}$ in the SoCAB and SJV, respectively. The contribution of the OVOCs to the OHR$_{VOC}$ was among the highest compared to the OHR$_{OVOC}$ measurements in other parts of the world (Table S2). In particular, the percentage of OVOC contribution was substantially higher than that in mainland China, Seoul (South Korea), and Mexico City (Mexico) (Table S2). HCHO was the most

important OVOC in terms of OHR, which was responsible for 45% and 36% of the OHR$_{OVOC}$ in the SoCAB and SJV, respectively. This percentage translates to HCHO accounting for ~30% of the OHR$_{VOC}$ in both basins. Acetaldehyde contributed 21% of the OHR$_{OVOC}$ and ~15% of the OHR$_{VOC}$ in both basins. Since ambient measurements of HCHO and acetaldehyde are scant compared to many other VOCs (e.g., aromatics), much remains to be learned on the sources and fate of these species. Methanol contributed 8% and 15% of the OHR$_{OVOC}$ in the SoCAB and SJV, respectively, while the ethanol

contribution to the OHR$_{OVOC}$ was ~7% in both basins. The sum of HCHO, acetaldehyde, methanol, and ethanol accounted for 51% and 69% of the OHR$_{VOC}$ in the SoCAB and SJV, respectively. Our measurements support the modeling results showing that 30%−50% of OHR$_{VOC}$ in California is due to aldehydes and other oxygenated species (Steiner et al., 2008).

Although BVOCs were only responsible for < 2% of the total measured VOC mixing ratio, they contributed 21% and 6% of

the OHR$_{VOC}$ in the SOCAB and SJV, respectively. Therefore it is important that the reactivity of BVOCs is well represented in photochemical models, especially in the SoCAB. The relatively high OHR$_{BVOC}$ is due to the high chemical reactivity of the BVOCs, i.e., the reaction rate constants of BVOCs with OH radicals are generally 100 times greater than those of alkanes (Table 2). Among the BVOCs, isoprene accounted for 59% and 44% of the OHR$_{BVOC}$ in the SoCAB and SJV, respectively.


The isoprene oxidation products MAC and MVK, together, accounted for ~45% of $OHR_{BVOC}$ in both regions. The

contribution of monoterpenes to $OHR_{BVOC}$ was < 10% in both basins. This should be considered as a lower limit as a large fraction of monoterpenes may have reacted away by the time they were measured from the aircraft. The dominance of isoprene and its oxidation products in $OHR_{BVOC}$ is consistent with the high isoprene emissions from the oak woodlands throughout the foothills of the Sierra Nevada mountains and near the northern boundaries of the SoCAB (Fig. S5a) (Arey et al., 1995; Benjamin et al., 1997; Misztal et al., 2014).

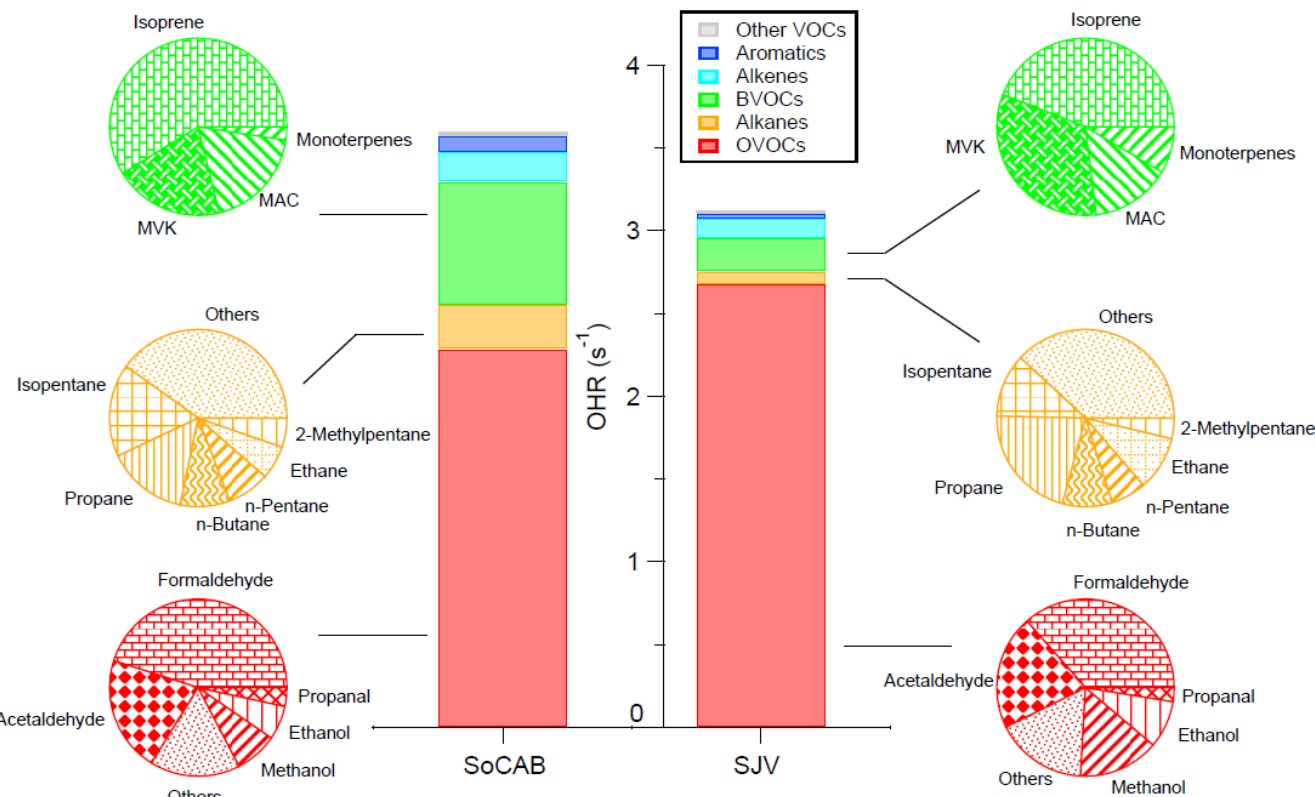

**Figure 6:** The measured OHR of VOCs in the SoCAB and SJV.


In contrast, the non-$CH_4$ alkanes only contributed 7% and 3% of the $OHR_{VOC}$ in the SoCAB and SJV, respectively, despite their greater atmospheric abundances. The sum of $OHR_{ethane}$ and $OHR_{propane}$ accounted for 21% and 31% of the $OHR_{alkanes}$ in

the SoCAB and SJV, respectively. The substantially smaller contribution of alkanes to $OHR_{VOC}$ compared to their contribution to the total VOC mixing ratio is due to their low chemical reactivity. On the other hand, the longer-chain alkanes dominated $OHR_{alkanes}$. Alkenes, aromatics, and other uncategorized VOCs (the "Other VOCs" in Table 2), together, accounted for < 10% of the $OHR_{VOC}$ in both basins.



### 3.5 Spatial variability of OHR$_{VOC}$

The OHR$_{VOC}$ showed distinct frequency distributions in the SoCAB and SJV (Fig. 7). The OHR$_{VOC}$ in the SoCAB spanned from 0 to 8 s$^{-1}$, with the highest occurrence frequency at 0−1 s$^{-1}$ and lowest at 7−8 s$^{-1}$. Higher OHR$_{VOC}$ occurred in more inland regions (Fig. 1), which was similar to the spatial distribution of O$_3$. This is likely due to the accumulation of air pollutants in downwind locations as they were transported from west to east during the day in the SoCAB (Carreras-Sospedra et al., 2006). In contrast, the OHR$_{VOC}$ in the SJV was narrowly distributed and centered at 1−3 s$^{-1}$. In addition, the

BVOCs in the SoCAB showed increasing contribution to the OHR$_{VOC}$ as the OHR$_{VOC}$ increased (the pie charts in Fig. 7a). The BVOCs contribution reached up to 20-30% when the OHR$_{VOC}$ was above 4 s$^{-1}$ near the northern boundary of the SoCAB (Fig. 1, Fig. S5b). This result suggests that BVOCs are likely key O$_3$ precursors during high O$_3$ episodes in the SoCAB. A similar result was not observed in the SJV, i.e., the OVOCs dominated the OHR$_{VOC}$ across the OHR$_{VOC}$ range. The minor role of the BVOCs in O$_3$ formation in the SJV was likely because of the daytime up-valley winds that prevented the

entrainment of the BVOCs emitted over the Sierra Nevada mountains into the valley (Zhong et al., 2004).

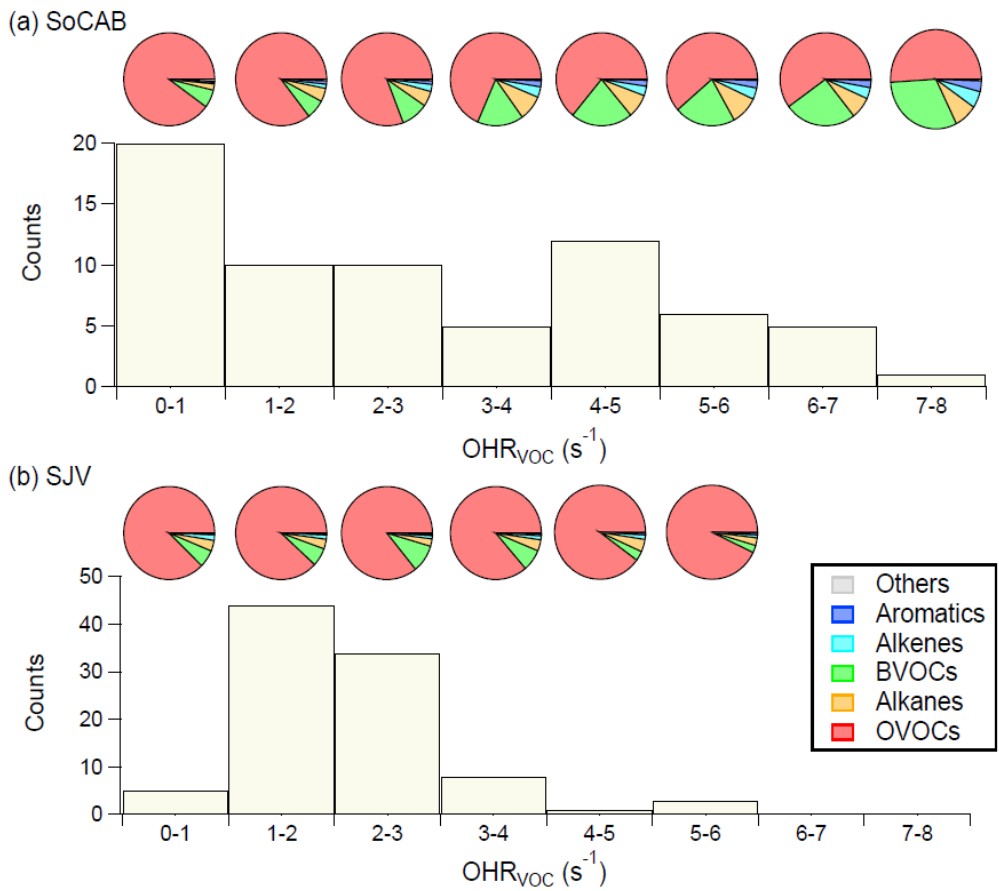

**Figure 7:** Frequency distribution of OHR$_{VOC}$ in the (a) SoCAB and (b) SJV. Also shown above each 1-s$^{-1}$ binned OHR$_{VOC}$ range is a pie chart presenting the average contribution to the OHR$_{VOC}$ by VOC categories (shown in the legend).



## 4 Conclusions and implications

In this work, we have updated the mixing ratios and OHR of a wide range of speciaed VOCs in the SoCAB and SJV of California. The spatially resolved data set and analyses can be used to evaluate the performance of photochemical models. Our measurements suggest that OVOCs are an important chemical class contributing to the OHR in California. Specifically, HCHO, acetaldehyde, methanol, and ethanol are the key OVOC species, the sum of which accounted for ~50%−70% of the calculated $OHR_{VOC}$ in the SoCAB and SJV. The observation is consistent with previous field and modeling studies showing that OVOCs play critical roles in atmospheric chemistry (Lou et al., 2010; Wu et al., 2020). OVOCs have also been suggested to be responsible for a major fraction of the "missing OH reactivity" (Dolgorouky et al., 2012; Karl et al., 2009; Lou et al., 2010), i.e., the difference between directly-measured $OHR_{TOTAL}$ and the sum of the calculated OHR from individually-measured species. However, quantifying OVOCs is still challenging and measurements of their ambient mixing ratios are still lacking in many regions. Our work reinforces that elucidating the chemical composition of OVOCs and their sources will advance our capability to predict $O_3$ abundance and develop $O_3$ reduction strategies.

Biogenic emissions represent a significant source of VOCs for $O_3$ formation, especially in the SoCAB. On average 21% and 6% of the $OHR_{VOC}$ are contributed by primary BVOCs in the SoCAB and SJV, respectively, assuming that isoprene is solely from biogenic sources. The contribution of BVOCs to $OHR_{VOC}$ was greatest near the northern edge of the SoCAB. This is consistent with the measurements in Pasadena, California, during the CalNex campaign, where isoprene was the single largest contributor to the $OHR_{VOC}$ (Heald et al., 2020). The importance of BVOCs as a contributor to OHR has also been reported for other urban environments, such as London, United Kingdom (Whalley et al., 2016), Beijing, China (Mo et al., 2018), and Seoul, South Korea (Kim et al., 2018). The $OHR_{BVOC}$ reported in this study should be considered as an underestimation, since a fraction of HCHO may be produced from the oxidation of BVOCs (Choi et al., 2010). Source apportionment of HCHO on the regional scale is thus critical to improving our future understanding of the contribution of biogenic emissions to $O_3$ production. Besides reacting with the OH radical, HCHO is also a source of $HO_x$ radicals ($HO_x$ = $HO + HO_2 + RO_2$) via photolysis. Reducing HCHO has the added benefit of reducing the production rate of $HO_x$, thereby lowering the production rate of $O_3$ (Pusede and Cohen, 2012).

As policies continue to curb anthropogenic emissions, BVOCs will continue to become increasingly important in atmospheric chemistry and $O_3$ formation (Gu et al., 2021). The temperature rise due to a warming climate will likely further enhance the emissions of BVOCs in the immediate future. Since BVOCs are highly reactive, a small increase in BVOCs will disproportionately enhance their contribution to the total OHR. This effect makes BVOCs even more critical during peak $O_3$ events that may lead to $O_3$ exceedance. Our current knowledge of the role of BVOCs in the $O_3$ formation in the South Coast region is still highly uncertain, e.g., recent photochemical modeling studies suggest a significant underestimation of biogenic



emissions in the SoCAB (Cai et al., 2019). As the contribution of BVOCs to $O_3$ production may represent a substantial fraction of the background $O_3$ that determines how stringent the $O_3$ standards will be, a more precise understanding of the magnitude and timing of BVOCs in $O_3$ formation is warranted. Such evaluation must be carefully paired with our evolving

understanding of the biospheric feedback in response to changing climate to improve mitigation and adaptation actions as well as future air quality management planning.

**Data Availability**

The data used in this study can be downloaded from the NASA FIREX-AQ data archive https://www-air.larc.nasa.gov/cgi-bin/ArcView/firexaq.

**Author contributions**

BB, RSH, AF, JP, SM, MS, AL, JBG, GIG, CW, ECA, AJH, IB, JW, PW, DR, and DB conducted the aircraft measurements. SL analyzed the data and wrote the manuscript. BB, RSH, AF, JP, MC, JBG, GIG, CW, TK, and MF provided constructive comments and suggestions.

**Competing interests**

The authors declare that they have no conflict of interest.

**Acknowledgements**

The authors acknowledge the funding from California Air Resources Board (contract #RD19014) to support the aircraft measurements. The HR-ToF-GC/MS measurement was supported by the National Center for Atmospheric Research, which is a major facility sponsored by the National Science Foundation under Cooperative Agreement No. 1852977. E.C.A.,

R.S.H., and A.J.H. were also funded in part by NASA Award No. 80NSSC18K0633. We thank the FIREX-AQ team, the coordinators, NASA, and NOAA for making this data collection possible with the DC-8 research aircraft. We thank Thomas Ryerson for providing the $O_3$ and $NO_x$ data, Glenn Diskin for providing the CO and $CH_4$ data, Paul Wennberg, Lu Xu, Krystal Vasquez, Hannah Allen, and John Crounse for providing the phenol and hydrogen cyanide data, Patrick R. Veres and J. Andrew Neuman for providing the NOAA CIMS data, and Gregory Huey for providing the Georgia Tech CIMS data. We

are grateful to Jeremy Avise and Chenxia Cai for their technical review and logistical coordination.



**Disclaimer**

The statements and conclusions in this manuscript are those of the authors and do not represent the official views of the California Air Resources Board.

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
