# Peer review of "Composition and Reactivity of Volatile Organic Compounds in the South Coast Air Basin and San Joaquin Valley of California"

_Atmospheric Chemistry and Physics, 2022_

## Author Response (AR1)

We appreciate the reviewers' comments, which have greatly helped improve the quality of the paper. To guide the review process, we have copied the comments in black text. Our responses are in regular blue font. The page and line numbers in this document refer to the originally submitted manuscript. We have responded to all reviewers' comments and submitted the revised manuscript with track changes.

**Reviewer #1:**

This study contrasts airborne VOC concentrations and reactivity in the SJV and SOCAB regions of California with a focus on the implications for $O_3$ in the region. The study is straight-forward and the manuscript was a pleasure to read – well organized and clear. I have only a few comments and suggestions.

Major Comments:

1. Rather than "total" OHR, I strongly suggest that the authors use the term "calculated", as in for example, Thames et al, ACP, 2020. The manuscript does not include OHR measurements and referring to "total measured OHR" as in the abstract (line 20) could be misleading.

    In the revised manuscript, we have used the term "calculated" as suggested. We have replaced "OHR" with "cOHR" (Heald et al. 2020) throughout the manuscript and the SI to indicate the OHR was calculated.

2. This study is based on only two flights. It raises the question of representativeness of the measurements shown here. I suggest that the authors discuss this directly in the conclusions and also consider softening some statements, particularly those that rely on the distribution of values. For example, line 140 "$O_3$ in the SoCAB may be more"

    We have added discussions of the representativeness of the two flights in the method section, where we can have more detailed discussions. We have compared the diurnal profiles of the air pollutants and meteorological parameters between the measurement days with the days during July–September 2019 using data from the surface air monitoring network. The comparison shows that measurement days are representative of typical summer days in the SoCAB and SJV. A figure (Fig. 1 in this document) has been added to the SI to support the discussion.

    We have added a paragraph after the 1st paragraph on Page 3: "To demonstrate the representativeness of the two flights, we synthesized data from the surface monitoring network (https://aqs.epa.gov/aqsweb/airdata/download_files.html) in the SoCAB and SJV for the summer of 2019. We then compared the diurnal profiles of the air pollutants and meteorological parameters between the two measurement days with the days during July–September, 2019. Examples are shown for Los Angeles County in the SoCAB and Kern County in the SJV (Fig. S2). The $O_3$ and air temperature profiles from the two measurement

days overlapped with those from July–September. These comparisons show that the measurement days are representative of typical summer days in the SoCAB and SJV."

We have also softened some statements as suggested. These include:
- Line 140, "O₃ in the SoCAB is more" → "O₃ in the SoCAB may be more"
- Line 174, "likely led to" → "may lead to"
- Line 185, "was likely the dominant" → "may be the dominant"

[Figure]

Fig. 1 Diurnal profiles of air temperature and O₃ for Los Angeles (LA) County in the SoCAB and Kern County in the SJV. The data are retrieved from the surface monitoring network (https://aqs.epa.gov/aqsweb/airdata/download_files.html). The box plots represent data of July−September 2019. The solid points represent data from July 22 (red) and September 5 (blue) of 2019.

3. The authors provide some contrast with previous studies, including CalNex (lines 259-261), but it would be informative to include even more discussion of how the SoCAB measurements in 2019 compare to the conditions in 2010 (and the earlier surface measurements in California listed in Table S1). In particular, it would be useful to discuss the relative decreases in VOCs and NOx, the impact on OHR, and any implications for the O₃ production regime. There were

also a couple of specific interesting differences from CalNex (as described in Heald et al., 2020) that would be useful to discuss the potential causes of: (a) ethanol:methanol > 2 in CalNex whereas this study reports that methanol concentrations exceed ethanol in the SoCAB in 2019 (b) limonene is below DL in this study (but comparable to methacrolein in 2010) – this seems interesting in light of the emphasis that this study places on the need to understand BVOC impacts on $O_3$.

We provided brief comparisons with previous studies because those studies are from ground measurements that are closer to the emission sources compared to ours. Vertical measurements have shown a fast concentration decrease from the surface to a few hundred meters above the ground. Therefore, it is not straightforward to compare flight data with ground data. We intended to avoid making detailed comparisons as the differences may arise from differences in aging and dilution rather than emissions, i.e., more aged and diluted air was measured during this study. This may explain why limonene was below the detection limit in this study (reacted away) whereas limonene was measurable during the CalNex surface measurements and the ethanol-to-methanol ratio was much lower in this study (ethanol is 3 times more reactive than methanol). In addition, comparing spatial measurements that cover a large region with stationary measurements at one location may be misleading. We have noted the caveat of the flight vs ground comparisons (Line 239-240) and added more caveats in response to comment #10 of Reviewer 1. Although detailed comparisons with previous surface measurements were not made, we want to put our study into a global context, so we summarized previous studies in Table S1 and Table S2.

Minor Comments/Corrections:

1. Line 25: "BVOCs were important"

   This is corrected.

2. Line 42: "relatively high O3 background" – relative to what? I suggest that the authors modify to provide specific numbers (i.e. Fig 1 of Parrish et al. indicates 20-40 ppb, varying by season). Parrish et al., 2017 also show that the background is decreasing which would be worth stating here as well.

   We have provided the specific numbers for background $O_3$. We have also stated that the background $O_3$ has been slowly decreasing and provided the decreasing rate retrieved from a new reference.

   Line 42, the revised text reads: "For example, Parrish et al. (2017) showed that the observational-based background ODV for southern California was 62±1.9 ppb, which was comparable to the modeled background ODV of 45−65 ppb with an outlier of ~92 ppb. This background ODV has decreased very slowly at ~1 ppb decade$^{-1}$ since the mid-2000s (Parrish et al., 2021)."

3. Figure 1: it's not really possible to distinguish the red and blue lines for flight 1 and 2 given the data overplotted. Perhaps for clarity the authors might consider removing those lines in Figure 1 and adding a figure in the SI that more clearly shows the two flight tracks.

We have added a figure (Fig. S1) to the SI that only shows the two flight tracks as suggested. Fig. S1 is cited after "We used data from the two California flights" in Line 61-62.

4. Table 1, entry #1: first name is "Thomas"

This is corrected.

5. Line 88-89: why is the slope on Fig S1 1.07? One would not expect mean values to be modified by merging to a different time base. Does this also reflect the filtering? If so, perhaps clarify this in the text.

The slope is slightly different than 1 because of data merging. The techniques we used have different sampling durations. When they were merged to the discrete WAS samples, some of the original samples were "lost". For example, when we merged the continuous 1-s ToF-CIMS data to the WAS sample time stamps (~40 s), the ToF-CIMS measurements obtained during the WAS sample intervals were not used. Therefore the merged data set is a subset of the original data set, resulting in a slope of 1.07. The comparison is to make sure that the subset of the original data set (merged data) can represent the original data set.

We have added "(a subset of the original data sets)" after "merged data set" in Line 86-87 to clarify this point.

6. Line 93: were the higher altitude points also removed from Fig 1? If so, I suggest stating that here.

Yes, Fig. 1 shows the final samples used for data analysis.

We have added "These samples are shown in Fig. 1." after "final data set for this study has 69 and 95 samples for the SoCAB and SJV, respectively." in Line 94.

7. Table 2: either footnote c or the top row should indicate that kOH is given at 298K. Footnote c should also state that references for reaction rates are given in the text, as readers may look for those here.

We have added "$k_{OH}$ values are given at 298 K. References for the $k_{OH}$ values are provided in the text." to footnote c as suggested.

8. Figure 4 discussion: The NOx OHR vs O3 relationship is very different between SJV and SoCAB – given that this is shown on the figure, it bears some discussion in the text.

The $OHR_{NOx}$ vs $O_3$ relationship between the SoCAB and SJV appears to be similar when they are plotted together (Fig. 2 in this document), except that the $OHR_{NOx}$ for SoCAB spans a larger range and has more scatters compared with $OHR_{NOx}$ for SJV. In light of Reviewer 2's comments, we have moved this figure to the SI since it is not discussed a lot.

[Figure]

Fig. 2 Scatter plot of $OHR_{NOx}$ versus $O_3$.

9. Line 253: you showed in Figure 3 that "the overall OHR in the SoCAB was 30% higher than in the SJV", so perhaps "confirming as in Fig. 3" rather than "suggesting" would be appropriate here.

We have replaced "suggesting" with "confirming" as suggested.

10. Line 269 vs lines 239-240: Earlier in the text the authors suggest that it's difficult to compare ground and airborne, but then here in line 269 the airborne measurements are compared to two ground sites and a model(!) in other regions of the world. If the authors stand by their statement on 239-240, then perhaps a caveat is needed here.

We have added a caveat after the comparison in Line 269: "We note that part of the difference may be due to the different study approaches, e.g., ground vs flight measurements."

11. Line 276-277: The Steiner et al. (2008) study is representative of conditions over a decade prior to this study, and clearly emissions have changed substantially in California over that time period (as shown here in comparison with previous work), so I suggest re-rephrasing "our measurements support" to "our measurements are consistent with…from over a decade prior."

The sentence has been rephrased as suggested.

12. Line 346: says "studies" but only one citation provided. Modify for consistency.

The sentence has been modified to read "a recent photochemical modeling study suggests…"

**Reviewer #2:**

Review Summary:

Liu et al present atmospheric composition measurements collected during two flights associated with the FIREX-AQ campaign in 2019. They analyzed VOCs, CO, NOx, and ozone from measurements collected below 1.2 km altitude. They estimated total OH reactivity including contributions to OH reactivity by different chemical constituents. They found that VOCs accounted for most OH reactivity, and oxygenated VOCs, in particular, accounted for more than 60% of the OH reactivity attributed to VOCs. Biogenic VOCs comprised a minor portion of the total VOC mixing ratio but accounted for 21% of the OH reactivity from VOCs in the South Coast Air Basin. Biogenic VOCs contributed to less of the OH reactivity in the San Joaquin Valley on these flight days. A steeper gradient in OH reactivity was observed in the South Coast Air Basin than the San Joaquin Valley, the latter of which was more homogeneously distributed. The data-set is unique, valuable to the scientific community, and a good fit for the journal. The analysis and context of the measurements could be improved with some minor modifications that I describe below.

General Comments:

The authors should comment on expected temporal variability throughout the year to provide some context for how representative these two flight days were. Would you expect BVOC and OVOC emissions to be higher or lower in the winter and spring? And why? Many of the plants in the region are drought deciduous and go dormant in the summer and fall. In the San Joaquin Valley, what is the context for agricultural activity on these two days? Are there times of year when the BVOCs from ag plants might have a stronger contribution to OH reactivity in this area?

BVOC emissions are known to be temperature-dependent (Guenther et al., 1993). Isoprene, the dominant BVOC, is mainly emitted from oak trees (Misztal et al., 2014) that enter dormancy by late November. Therefore we expect BVOC emissions to be higher in the summer compared to other seasons. OVOCs have both primary and secondary sources. The abundance of the secondary OVOCs is expected to be higher in the summer due to higher photoactivity. The primary OVOCs may be more abundant in the colder seasons because of relatively lower PBL height and higher frequency of stagnant air, but this needs to be confirmed by long-term OVOC measurements in California (currently unavailable to the best of our knowledge).

The two measurement days are representative of typical summer days in the SoCAB and SJV (please refer to the response to Reviewer 1's major comment #2). Fares et al. (2012) examined the seasonal cycle of the BVOC flux by conducting flux measurements in Exeter in the SJV for a year. They found that the highest BVOC emissions were observed during the flowering period in the spring. So the agricultural plants might have a stronger contribution to OH reactivity in the springtime. However, there is very limited data on VOC emissions from agriculture in California. The influence of agricultural activities on BVOCs and agriculture plant VOC emissions requires further studies on different agricultural plant types at different seasons in California.

We have added "It is expected that the summer period features high $O_3$ pollution as well as high BVOC emissions and secondary OVOC productions." after Line 67.

The conclusions would be better supported with some simple box modeling to estimate how much of the VOCs could have reacted away by the time the air parcel reached the aircraft. The authors

mention this in a qualitative sense, but knowing the reaction rate constants, assuming some oxidant concentration, and estimating the air parcel age would allow them to be more quantitative in this assessment. Even some range of values would be helpful. Has >80% of the BVOCs reacted away already? <20%? They state that the VOCs have undergone photochemical processing for minutes to several hours, but this could be translated into a more meaningful estimate of how this equates to estimated percent loss of VOCs vs OVOCs. You could even estimate how much of the HCHO was contributed by BVOC oxidation vs OVOC oxidation, etc.

In light of the Reviewer's comments, we have estimated the photochemical age of the samples using the $NO_x$ and $NO_y$ measurements. Assuming that the ambient [OH] is 4 x $10^6$ molecules $cm^{-3}$, the estimated photochemical age of the sampled air was approximately 12 hours on average (Cappa et al., 2012). This means a compound with $k_{OH}$ of $10^{-11}$ $cm^{-3}$ $molecules^{-1}$ $s^{-1}$ had ~80% of its mass reacted away before it was sampled. Compounds with higher rate coefficients will have more loss. For example, most of the BVOCs likely had reacted away before they were sampled.

We have revised Lines 99-100 to read: "Using the $NO_x$ and $NO_y$ measurements and assuming that the ambient [OH] was 4 x $10^6$ molecules $cm^{-3}$, the photochemical age of the samples was estimated to be approximately 12 hours on average (Cappa et al., 2012). This means a species with an OH rate coefficient of $10^{-11}$ $cm^{-3}$ $molecules^{-1}$ $s^{-1}$ had lost ~80% of its mass before they were sampled by the aircraft. Compounds with higher rate coefficients (e.g., the BVOCs) had more loss. The statistics of the observed mixing ratios of the gas-phase species and their OH rate coefficients are summarized in Table 2."

Specific Comments

1. The authors mention they only included measurements collected below 1.2 km. Can they clarify the PBL height? Are these within the PBL or in the free troposphere?

   PBL height was not measured during our measurement. Previous observations have shown that the daily maximum PBL height (in the afternoon) was approximately 1–1.5 km in the SoCAB and 0.5–1 km in the SJV (Cui et al., 2019; Bianco et al., 2011). Therefore, the samples were most likely collected within the PBL.

   Line 98, we have added "Observations have shown that the daily maximum planetary boundary layer (PBL) height was approximately 1–1.5 km in the SoCAB and 0.5–1 km in the SJV (Cui et al., 2019; Bianco et al., 2011). Therefore the samples were most likely collected within the PBL."

2. Table 2 provides incredibly detailed measurement information, but it's unclear to me why it is in the methods section instead of the results. Aren't these results?

   We have moved Table 2 to the "Results and discussion" section as suggested.

3. It's unclear what the main takeaway is from Figure 4. Can the authors please strengthen the discussion of this figure, better highlight its relevance, or remove it if that is not possible? As far as I can tell, the figure isn't referenced anywhere in the text directly.

We agree that Fig. 4 is not discussed a lot (only referenced in Line 238). We have moved Fig. 4 to the SI.

4. On page 16, the authors make comparisons to OVOC contributions reported in mainland China, Seoul, and Mexico City. Can the authors clarify that these were also flight measurements at a similar altitude? Or were these surface measurements? It would be helpful to know to provide context for the comparison.

We have clarified the methods of the measurements reported in other regions in the text. The sentence was revised to read: "In particular, the percentage of OVOC contribution was substantially higher than that in mainland China (calculated from emission inventories), Seoul (South Korea, surface measurement), and Mexico City (Mexico, surface measurement) (Table S2)." The measurement platform information for these regions is also available in Table S2.

References

Bianco, L., Djalalova, I., King, C., and Wilczak, J.: Diurnal evolution and annual variability of boundary-layer height and its correlation to other meteorological variables in California's Central Valley, Boundary-layer meteorology, 140, 491-511, 2011.

Cappa, C. D., Onasch, T. B., Massoli, P., Worsnop, D. R., Bates, T. S., Cross, E. S., Davidovits, P., Hakala, J., Hayden, K. L., and Jobson, B. T.: Radiative absorption enhancements due to the mixing state of atmospheric black carbon, Science, 337, 1078-1081, 2012.

Cui, Y. Y., Vijayan, A., Falk, M., Hsu, Y.-K., Yin, D., Chen, X. M., Zhao, Z., Avise, J., Chen, Y., and Verhulst, K.: A multiplatform inversion estimation of statewide and regional methane emissions in California during 2014–2016, Environmental science & technology, 53, 9636-9645, 2019.

Fares, S., Park, J.-H., Gentner, D., Weber, R., Ormeño, E., Karlik, J., and Goldstein, A.: Seasonal cycles of biogenic volatile organic compound fluxes and concentrations in a California citrus orchard, Atmospheric Chemistry and Physics, 12, 9865-9880, 2012.

Guenther, A. B., Zimmerman, P. R., Harley, P. C., Monson, R. K., and Fall, R.: Isoprene and monoterpene emission rate variability: model evaluations and sensitivity analyses, Journal of Geophysical Research: Atmospheres, 98, 12609-12617, 1993.

Misztal, P., Karl, T., Weber, R., Jonsson, H., Guenther, A. B., and Goldstein, A. H.: Airborne flux measurements of biogenic isoprene over California, Atmospheric Chemistry and Physics, 14, 10631-10647, 2014.

Parrish, D. D., Derwent, R. G., and Faloona, I. C.: Long-term baseline ozone changes in the Western US: A Synthesis of Analyses, Journal of the Air & Waste Management Association, 71, 1397-1406, 2021.

Parrish, D. D., Young, L. M., Newman, M. H., Aikin, K. C., and Ryerson, T. B.: Ozone design values in southern California's air basins: Temporal evolution and U.S. background contribution, Journal of Geophysical Research: Atmospheres, 122, 11,166-111,182, 10.1002/2016jd026329, 2017.